# Effect of Varying Levels of Hempseed Meal Supplementation on Humoral and Cell-Mediated Immune Responses of Goats

**DOI:** 10.3390/ani11102764

**Published:** 2021-09-22

**Authors:** Frank Abrahamsen, Gopal Reddy, Woubit Abebe, Nar Gurung

**Affiliations:** 1College of Agriculture, Environment and Nutrition Sciences, Tuskegee University, Tuskegee, AL 36088, USA; fabrahamsen@tuskegee.edu; 2College of Veterinary Medicine, Tuskegee University, Tuskegee, AL 36088, USA; wabdela@tuskegee.edu

**Keywords:** goat, cell-mediated, antibody, industrial hemp

## Abstract

**Simple Summary:**

Hempseed meal (HSM) is a byproduct of hemp oil production and is high in protein, fiber, and fat. With hempseed having an ideal omega 6:3 ratios for human health, a similar ratio is observed in HSM. Currently, HSM is not approved for use in animal feed as there are several safety concerns for both the animal and consumer. In this study, we evaluated the effect of HSM supplementation for goats on their humoral, cell-mediated immune responses and select cytokine expression and serum concentration. Supplementation of HSM improved the cell mediated immune response but decreased the antibody response in goats. Including HSM in the diet of goats could improve the cell-mediated immune response.

**Abstract:**

The objective of this study was to evaluate the effect of varying levels of hempseed meal supplementation on antibody and cell-mediated immune responses, as well as the expression of some of the important immunoregulatory cytokines. Treatments consisted of hempseed meal supplementation at 0 (control), 10, 20, and 30% of the total diet. Goats were randomly assigned to one of the four treatments *n* = 10. Cell-mediated immune response was evaluated on day 59 of the feeding period by measuring skinfold thickness at 24 h following intradermal injection of phytohemagglutinin. A significant increase in skinfold thickness was observed with increasing levels of supplementation as compared to that of the control group. Serum antibody titers to chicken ovalbumin were not significantly different between treatment groups. Cytokine concentrations of IL-6 increased linearly with increasing level of supplementation (*p* < 0.05), contrarily to the linear decrease that was observed for TNF-α (*p* < 0.05). Although IL-2 tended to increase with the 10 and 30% levels of supplementation (*p* < 0.07), the result was not significant, and no significant differences were obtained with respect to IL-4 concentrations. Cytokine gene expression values measured by RT-PCR, however, demonstrated some significant differences. HSM supplementation had no significant effect on the expression of IL-2 or IL-6. However, significant differences were observed with the 30% supplementation for IL-4 and TNF-α as compared to that of the control group (*p* < 0.05). IL-4 was down regulated for the 10 and 20% treatment groups but was upregulated for the 30% treatment group. TNF-α was downregulated in the 10% but upregulated for the 20 and 30% treatment groups. No significant differences were observed for the serum cortisol concentration or white blood cell counts. These results suggested that hempseed meal supplementation may improve cell-mediated immune response while having no effect on antibody-mediated immune response. However, more research needs to be conducted to determine the most efficacious inclusion rate.

## 1. Introduction

Industrial hemp production in the United States is likely to increase in the future due to legislative changes to the Farm Bill [1]. Currently, several states established pilot programs allowing them to shape their legislation better and assess the viability of Industrial Hemp as a crop. Industrial hemp can be grown for oilseed and fiber production [2]. Oilseeds are produced mainly to meet the demand of niche health markets for humans. Hemp oil extraction is accomplished by either cold-pressing methods or by solvent-based extraction methods in some cases. The omega-6 to omega-3 ratio of hemp seed oil is considered to be optimal for human health [1]. After the oil is extracted, the residual meal also called hempseed meal (HSM) is high in protein, fiber, and fat, potentially making it an excellent feed for ruminants [3,4,5,6,7]. HSM consistently has a crude protein concentration between 30–35% on a dry matter basis [6,7]. The feeding value of HSM could play a significant role in determining the sustainability of industrial hemp in the United States.

The fatty acid composition of diets was long determined to have a significant effect on immune responses. Additionally, unsaturated fatty acids have antimicrobial properties, and the rumen microbes work to transform these fats into stearic acid [8]. Not all unsaturated fatty acids are converted in the rumen; escaped fatty acids have the potential to influence the composition of meat and other physiological processes in the animal. For years, scientists researched fatty acids and their immunomodulatory effects, as they give rise to inflammatory mediators such as prostaglandins and leukotrienes, and alter the membrane fluidity of immune cells [9,10]. Additionally, trace amounts of CBD (cannabidiol) could potentially be found in the oil, which can also have anti-inflammatory effects [11,12,13]. To our knowledge, comprehensive studies describing the effects of different levels of HSM supplementation on the immune responses of animals are lacking. Studies are needed to find the optimum level of supplementation that has a beneficial effect on animal health and production while being cost-effective for the farmers (producers).

The objective of this study was to evaluate the effect of varying levels of HSM supplementation on cell- and antibody-mediated immune responses, including cytokine expression levels and serum cortisol levels.

## 2. Materials and Methods

### 2.1. Ethics

All handling, animal care, and sample collection procedures were approved by the Tuskegee University, Institutional Animal Care and Use Committee (TUACUC Protocol Request Number: R07-2019-5).

### 2.2. Animals, Experimental Design, and Feeding

Forty 4 to 5-month-old castrated Boer cross goats were randomly assigned to one of the 4 treatments of HSM supplementation diets at 0, 10, 20, and 30% levels (*n* = 10). The average initial body weight of all goats was measured and recorded at 25.63 ± 0.33 kg. Animals had ad libitum to treatment diets, and refused feed was weighed twice daily to determine feed intake. Long stem hay was provided at a rate of 0.23 kg per day to ensure proper rumen function. Additionally, all animals were provided *ad libitum* access to water.

### 2.3. Feed Manufacturing and Analysis

The complete diet consisted of timothy hay, soybean meal, meat maker 16:8 (goat premix), corn, molasses, and hempseed meal at varying rates (0, 10, 20, and 30%; Table 1), which were pelleted at Auburn University Poultry Feed Mill. This was done to ensure the goats would consume as much hempseed meal as possible by reducing the chance of selecting other portions of the diet.

A total of 350 bags were made, and during the bagging process, 0.23 kg of the complete feed was collected from every 5th bag and distributed homogenously. Eighty bags were prepared for each of the four treatment groups (0, 10, 20 and 30%). Pelleted samples were then shipped to Holmes Laboratory (Millersburg, OH, USA) for determination of nutrient composition. Analysis of dry matter, crude protein, acid detergent fiber, acid hydrolysis fat, ash, phosphorus, magnesium, potassium, sulfur, manganese, zinc, and iron were completed according to the methods described by the American Organization of Analytical Chemist [14]. Neutral detergent fiber content was determined utilizing an Ankom 2000, Fiber Analyzer (Ankom Technology, Macedonia, NY, USA) according to the manufacturer’s recommendations. Lignin concentration was determined according to methods described by the United States Department of Agriculture [15]. Fatty acid analysis was completed for each diet by Eurofin Microbiology Lab (Atlanta, GA, USA) according to the methods described by the American Organization of Analytical Chemists [14].

### 2.4. Blood Collection, White Blood Cell Counts, Gene Expression, and Serum Cytokine and Cortisol Concentration

Blood was collected via jugular venipuncture utilizing a 21-guage needle on day 60 of the feeding period for the analysis cortisol concentration, cytokine concentration, white blood cell counts, antibody response, and cytokine gene expression. Blood for serum cortisol and cytokine concentration was collected utilizing a VACUTTE Tube, 3.5 mL CAT Serum Separator/Clot Activator Tube (Greiner Bio-One, North America Inc., Monroe, NC, USA). Serum was separated and collected by centrifuging the samples at 2000× *g* for 30 min. Blood for gene expression and white blood cell count analysis was collected utilizing a 9 mL, VACUTTE Tube, K2 EDTA (Greiner Bio-One, North America Inc., Monroe, NC, USA). White blood cell counts were completed using an IDEXX, Procyte DX. Additionally, on day 60 of the feeding period, peripheral blood mononuclear cells (PBMC) were collected from whole blood according to the manufacturer’s recommendations, utilizing lymphocyte separation media (Sigma–Aldrich, St. Louis, MO, USA).

Cytokine concentrations in the serum samples for interleukins, IL-2, IL-4, IL-6, and tumor necrosis factor-α (TNF-α) were determined utilizing sandwich ELISA kits purchased from CUSABIO^®^ (Houston, TX, USA). Concentrations were determined according to the manufacturer’s recommendations.

Total RNA was extracted from the PBMCs utilizing Tri-Reagent according to the manufacturer’s recommendations (Sigma–Aldrich, St. Louis, MO, USA). Primers for the cytokines IL-2, IL-4, IL-6, and TNF-α (Table 2) were designed utilizing Vector NTI Software (Thermofisher Scientific, Carlsbad, CA, USA) and purchased from Integrated DNA technologies (Coralville, IA, USA). Murine Leukemia Virus Reverse (MLV) transcriptase was the enzyme utilized for reverse transcription of the total RNA from peripheral mononuclear blood cells. Βeta-actin was used as an endogenous housekeeping gene control. StepOne Plus (Life Technologies, Inc., Carlsbad, CA, USA) real-time PCR system was used for all the PCR assays with SYBR^®^ Green Quantitative RT-qPCR Kit (Sigma–Aldrich). cDNA synthesis was performed with an initial annealing and polymerization temperature of 55 °C and a second step of enzyme deactivation at 95 °C for 2 min. The PCR reactions consisted of 10 µL of 1X hot start Taq DNA Polymerase, 1 µL of 600 nM forward and reverse primers, 2 µL of 200 ng/µL RNA template, 0.1 µL M-MLV, and 6 µL of molecular biology water to a final volume of 20 µL. The RT-PCR conditions were 40 cycles of denaturation at 95 °C for 15 s and annealing and extension steps at 52 °C for 1 min. Fluorescence data were collected on the down-ramp to the annealing and extension step and during the annealing and extension holding step. Melting curve analysis was completed with a starting temperature of 52 °C and a final temperature of 95 °C. Only amplification that appeared before the 35th cycle was considered in the analysis. Relative quantification values (2^−ΔΔct^) were calculated utilizing the StepONE Plus data analysis software and represent fold change compared to that of the control.

### 2.5. Cell Mediated and Humoral Immunity Analysis

On day 58 of the feeding period, cell-mediated immunity was assessed by intradermal injection of 0.5 mL of 2 mg/mL phytohemagglutinins (PHA; Sigma–Aldrich, St. Louis, MO, USA) dissolved in 0.9% sterile saline solution. The injection was performed specifically at the left caudal fold of each goat. Before the intradermal injection, the area was thoroughly cleaned with 70% isopropyl alcohol, and the initial measurement was taken. Skin thickness was measured again 24 h post intradermal injection to determine the change in skin thickness.

Antibody responses were assessed on day 60 of the experiment via a chicken ovalbumin challenge given at day 46. Chicken ovalbumin (cOVA; Sigma–Aldrich, St. Louis, MO, USA) was prepared at a concentration of 5 mg/mL with sterile saline solution. Alhydrogel (Invivogen, SanDiego, CA, USA) was utilized as an adjuvant and was mixed at a 1:1 ratio with the previously reconstituted antigen solution. Each animal received a subcutaneous injection of 1 mL of the preparation on the left-side of the neck. Antibody response was assessed 14 days’ postinjection via indirect ELISA.

Antibody responses specific to cOVA were determined using serum that was collected on day 60 of the feeding period. Serum was separated as described above (Section 2.4) and stored at −20 °C until further analysis. A 96-well ELISA plate (Invivogen, SanDiego, CA, USA) was coated with 50 µL of a 20 µg/mL of cOVA in 100 mM bicarbonate/carbonate coating buffer and incubated overnight at 4 °C. Wells were then washed three times, each time filling the wells with 200 µL of 1X PBS. Plates were then blocked by adding 200 µL of 1% fetal bovine serum/PBS to each well and incubating it at room temperature for two hours. Next, 100 µL of the sample was added in triplicate at the following dilutions: 1:10, 1:20, 1:40, 1:80, 1:160, and were incubated at 4 °C overnight. Plates were then washed 4 times utilizing 1X PBS. Anti-goat IgG peroxidase-conjugated antibody (Jackson ImmunoResearch, West Grove, PA, USA) was then prepared at a 1:500 dilutions, and 100 µL of the conjugate was added per well and incubated at room temperature for two hours. The plates were again washed four times using 200 µL of 1X PBS per well. Chemiluminescent peroxidase (Sigma–Aldrich, St. Louis, MO, USA) was added to each well at a volume of 50 µL and allowed to incubate for 5 min at room temperature. At the end of the final incubation period, the contents of each well were transferred to a white opaque bottom cell culture plate (Falcon 96 well, white opaque bottom plates), and luminescence was read (Synergy 2 & Biotek, Winooski, VT, USA). The negative control was subtracted from luminescence values.

## 3. Results

### 3.1. Nutrient Composition and Fatty Acid Composition of Diets

#### 3.1.1. Nutrient Composition

The first hempseed meal, which was used in formulating the different diets, had a relatively high crude protein concentration (Table 3). Diets formulated in this study were formulated to be isonitrogenous, and each diet had a similar crude protein concentration; however, the crude protein concentration of the 30% diets was slightly elevated when compared to that of the three other diets (Table 3). Hempseed meal is relatively high in fiber and fat; with the increasing rate of supplementation, an increase was observed in both fiber and fat. Lignin also increased with the increasing rate of hempseed meal supplementation (Table 3). Calcium concentration decreased with the increasing level of hempseed meal supplementation, while phosphorus increased. Magnesium increased slightly with the increasing level of supplementation, while potassium followed a decreasing trend. Sulfur concentration remained constant despite the level of hempseed meal supplementation. The concentration of sodium decreased from the control to 10%, and then increased and remained steady for the remaining treatments. Copper concentration followed a quadratic relationship with a decrease from control to 10% but a slight increase from 10% to 20%, followed by a greater increase to 30%. Manganese concentration increased with the increasing level of hempseed meal supplementation while zinc decreased from control to 10% but increased for the remaining treatments. Iron concentration increased from the control to 10% treatment and remained the same for the 20% treatment, but it decreased to a level lower than that observed in the control for the 30% treatment.

#### 3.1.2. Fatty Acid Composition of Diets

Table 4 displays the fatty acid analysis and concentration conducted on pure hemp seed meal and the respective diets. With the increasing levels of HSM supplementation, we observed a consistent increase in the levels of linoleic acid, linolenic acids, and concentrations of total omega 3 and 6 isomers. In addition, both monounsaturated and polyunsaturated fatty acids increased with the increasing levels of HSM supplementation; this trend was also true for the total fats, such as triglyceride and total fatty acids.

Palmitic acid concentration slightly increased with the 10–20% treatment diets, followed by a slight decrease from the 20–30% treatment group. Stearic acid followed a more inconsistent pattern, as there were increases and decreases with the increasing level of supplementation. Vaccenic acid concentration was extremely low (i.e., <0.03) for the control and 10% treatments, while the other two treatments revealed slightly higher and measurable concentrations. Oleic acid concentration displayed an irregular pattern of increasing and decreasing between treatments.

### 3.2. White Blood Cell Counts, PHA Challenge, Anti-cOVA Antibody Production, and Cytokine Gene Expression

#### 3.2.1. White Blood Cell Counts

Increasing levels of HSM supplementation had no significant effect on total white blood cell counts as the counts were similar between the control group and treated groups. Similarly, there were no significant differences in white blood cell differential counts between HSM-supplemented and control goat groups (Table 5).

#### 3.2.2. Skinfold Thickness Changes Following Intradermal Injection of PHA

Thickness of the caudal fold was measured before and 24 h after intradermal injection of PHA on day 59 of the feeding period as a measurement of cell-mediated immunity. Skinfold thickness was higher in all HSM supplemented groups as compared to that of values recorded for control group goats. However, the 10% treatment group showed higher values compared to groups that were supplemented at 20 and 30% levels (Table 6). The differences between the control and the 10%, 20%, and 30% were determined to be significant (*p* ≤ 0.05).

#### 3.2.3. Serum Anti-cOVA Antibody Titers Determined by ELISA

All supplemented and control group goats demonstrated serum anti-cOVA antibodies in the blood samples collected on day 60, even at the highest level of serum dilution (1:160). Antibodies were detected at all serum dilutions (1:10, 1:20, 1:40, 1:80, and 1:160; Figure 1), and we observed significantly lower levels of antibody responses in all the three HSM-supplemented groups as compared to that of the control group goats (*p* < 0.05). The results of the antibody response at 1:160 dilutions from animals supplemented with 20 and 30% HSM were significantly (*p* < 0.05) different from that of those supplemented 10% HSM.

#### 3.2.4. Cytokine Concentration & Serum Cortisol Concentration

HSM at all supplemented levels had no effect on *IL-4* concentrations determined by ELISA in serum samples collected on day 60 of the feeding period (Table 7). Serum *IL-6* concentrations increased linearly (*p* = 0.01), while *TNF-α* decreased linearly (*p* = 0.01) with the increasing level of HSM supplementation (Table 7). Serum *IL-2* concentrations were higher in goats supplemented with HSM at 10% and 30% levels but were lower at 20% level compared to that of nonsupplemented control groups; however, these results were not statistically significant (*p* = 0.08; Table 7). Serum cortisol concentrations determined by ELISA in serum samples collected on day 60 remained relatively constant for all treatments demonstrating no significant differences (Figure 2).

#### 3.2.5. Cytokine mRNA Expression in Peripheral Blood Mononuclear Cell (PBMCs)

Expression of IL-2 and IL-6 were not significantly impacted by HSM supplementation (Table 8). However, compared to that of control groups, IL-4 expression was not significantly impacted in the PBMC of goats that received HSM supplementation at 10 and 20% levels. On the other hand, we observed an upregulation of IL-4 expression in the PBMCs of goats that were supplemented with 30% HSM as compared to that of all other treatments including the control group. Only the 30% supplemented group showed significantly higher value compared to that of the other treatments with respect to TNF-α.

## 4. Discussion

As part of the current research, we conducted a parallel study that evaluated the effect of HSM supplementation in animal performance. The observation is that as the level of HSM supplementation increased, there was a linear decrease in live weight gain, feed efficiency, and rumen fermentation. However, the inclusion of HSM at different proportions did not significantly impact the consumption of dry matter over the feeding period. Even though a linear decrease was observed in most of the animal performance parameters, animal performance in 30% treatment was still considered relatively good for growing meat goats.

### 4.1. Fatty Acid Composition of Diets

Hempseed was long touted for being an exceptional health supplement because of its ideal omega 6: omega 3 ratio [1]. Hempseed meal still contains less fat than the entire seed; however, it still provides a relatively high concentration of similar fatty acids to that of the whole seed. Essential fatty acids that are not transformed in the rumen escape to the circulation and can potentially influence the immune function of animals, as in monogastric animals. The omega 6:3 ratio of pure hempseed meal utilized in this study was 3.75. The highest omega 6:3 ratio was reported for the control group due to significant amounts of corn and soybean meal, which both contain higher amounts of omega 6 fatty acids. In this study, we noted that as the level of HSM increased, the omega 6:3 ratio decreased, implying higher levels of omega 3 in HSM. Further, the hempseed meal utilized in this project was higher in linoleic acid, an omega 6 fatty acid, which is considered more proinflammatory than any anti-inflammatory omega 3 fatty acids. Fatty acids themselves do not directly influence cytokine production; they are processed into prostaglandins and leukotrienes, which have the pro- and anti-inflammatory effects. Omega 6 fatty acids lead to the production of four-series of leukotrienes, which are responsible for chemotactically calling neutrophils to the site of infection [16]. Additionally, the four-series of leukotrienes can also recruit mast cell progenitors to the site of infection, which can trigger the production of some proinflammatory cytokines [17].

#### 4.1.1. White Blood Cell Counts 

Increasing levels of HSM supplementation had no significant effect on the total white blood cell counts or differential counts of white blood cell subsets. These results are somewhat similar to earlier reports of Iannaccone et al. [18], who supplemented 5% of the total diets with hempseed meal for lactating ewes and did not identify any significant difference with respect to complete white blood cell counts or the differential counts. Li et al. [19] investigated the impact of soybean oil and fish oil supplementation on total white blood cell counts of rats and did not find any significant differences between the two oils; however, a significant difference was identified in rats supplemented with lard.

#### 4.1.2. Skinfold Thickness Following Intradermal PHA Challenge

On day 58 of the feeding trial, cell-mediated immunity was assessed via phytohemagglutinin challenge. Skin thickness was calculated based on the difference between final and initial thickness 24-h postintradermal injection. All treatment means for HSM supplemented groups were significantly higher (p ≤ 0.05) than that of the control group. Skin thickness was greatest for the 10% treatment, while it was slightly less for the 20 and 30% HSM-supplemented groups; regardless, the values were significantly higher than that of the control groups. As shown later, these results positively correlated with expression levels of IL-2, a Th1 cytokine predominantly involved in augmenting cell-mediated immune responses, such as cytotoxicity against virus-infected cells. These data suggest that HSM supplementation may have a slight positive influence on enhancing cell-mediated immunity. In this study, as the inclusion rate of hempseed meal increased, we observed a decrease in the omega 6:3 ratio. Kiecolt–Glaser et al. [20] described that the effect of increasing levels of omega 3 fatty acids in the diet of humans had a positive impact on the cell-mediated immune response. Data of this study are also in line with the findings reported by Caroprese et al. [21], who reported that if there was any change in an immune response then it would be towards a Th1 response as there was no change in antibody response of dairy cattle supplemented with either fish oil or flaxseed. However, these results do not agree with report of Kumar et al. [16], which described how increasing the level of omega 3 in the diet would cause a shift to more of a Th2 response. Data of this study with different levels of HSM supplementation for goats suggest that decreasing levels of omega 6:3 ratio may have some beneficial effects on cell-mediated immunity, which may help goats to resist viral and other intracellular pathogens.

#### 4.1.3. Serum Anti-cOVA Antibody Responses

The results of this study demonstrated that supplementation of the hempseed meal at increasing levels may result in a slight decrease in specific antibody responses. Studies of Caroprese et al. [21], who supplemented dairy cattle with flaxseed, found no differences in antibody response to chicken ovalbumin. Caroprese et al. [22], who supplemented goats with linseed and fish oil, showed a decrease in antibody response and no change in the cell-mediated immune response compared to that of the control groups. Interestingly, both linseed and fish oil consist of more omega 3 fatty acids than that of the diet in the control. In the current study, the higher the omega 3 content of the diet revealed a decrease in the antibody-mediated responses; on the contrary, we observed an increased in cell-mediated immune response. We believe the diet in this study had a slowly increasing gradient of omega 3 fatty acids, and the amount after rumen transformation was possibly insufficient to alter antibody-mediated immune response, as seen with Caroprese et al. [22].

#### 4.1.4. Concentration of Select Cytokines & Expression

Jaudszus et al. [23] demonstrated with cell culture work that increasing the levels of omega 3 fatty acids resulted in less intracellular concentration of IL-2. These results were not seen in the present in-vivo study with goats that received increasing levels of omega 3 with increased levels of HSM supplementation. Furthermore, in the present study, increasing levels of HSM supplementation resulted in decreased levels of omega 6:3 ratio.

An apparent increase in IL-2 concentration was observed with the increasing level of hempseed supplementation, which could almost be described as a linear relationship except for result at 20% HSM supplementation. The difference observed for the 20% treatment group cannot be explained conclusively. The decrease observed in the concentration of IL-4 (Table 7) could potentially be explained because of the increasing availability of omega 3 fatty acids that can successfully be converted to DHA or EPA. As described by Jaudszus et al. [23], EPA had the potential to decrease the intracellular concentration of IL-4 in cell culture work. In the present study, a decrease was observed with hempseed meal supplementation at 10% and 20% levels; however, an increase was observed with respect to the 30% treatment. This difference could be attributed to the differences that were observed in feed intake.

Tumor necrosis factor-α concentration decreased with the increasing level of hempseed meal supplementation. Skuladottir et al. [24] showed that the supplementation of n-3 PUFAs increases the secretion of TNF-α from macrophages. However, Jaudszus et al. [23], showed that the intracellular concentration of TNF-α in Th cells is significantly reduced when cells were supplemented with DHA. The differences in these two studies could be explained by the difference in macrophages and Th cells. In our in-vivo study, we estimated serum concentrations, which more consistently agreed with the trend described by Skuladottir et al. [24]. IL-6 concentration increased with the increasing level of hempseed meal supplementation in a linear fashion. As the hempseed meal inclusion rate increased, the level of omega 3 fatty acids also increased, making them more available to the animal. They also found that omega 3 fatty acid supplementation in humans infected with HIV had the ability to reduce the plasma concentration of IL-6. In this experiment the amount of omega 3 fatty acids available to the animal increased with the increasing level of supplementation. However, there is still more omega 6 than omega 3 because of the nature of the hempseed meal. In the present study, the steady increase in IL-6 concentration is quite the opposite of what Coghill et al. [25] found in humans. These differences could potentially be explained by species differences and the fact that, in our study, we did not use omega 3 as a sole supplement.

IL-2 expression did not display any significant differences compared to that of the other cytokines. However, serum concentration did follow an increasing trend (*p* = 0.08), which is consistent with the results of gene expression for the 20 and 30% treatment groups. Additionally, the increase in expression at the 20 and 30% treatments and corresponding increase of IL-2 serum concentration possibly resulted in enhanced cell mediated immunity. Skin thickness as a result of intradermal injection of PHA in HSM supplemented animals was significantly greater than that of the control groups, further strengthening the beneficial effects of HSM supplementation on cell-mediated immune responses.

Serum IL-4 concentrations were slightly but nonsignificantly lower in all levels of supplementation. IL-4 expression in PBMC followed a similar trend except in the 30% HSM supplementation, where the expression was significantly upregulated. IL-4 has an important role in regulating antibody production, hematopoiesis and inflammation, and the development of effector T-cell responses. It is produced predominantly by mast cells, basophils, and a subset of T helper cells. In addition to regulating B cell growth and immunoglobulin secretion, IL-4 also induces the differentiation of naive CD4^+^ T cells into T_H_2 cells while simultaneously inhibiting the generation of T_H_1 cells, which secrete IL-2 and interferon- γ (IFN-γ) [26]. Decreased production of IL-4 may partly explain the decreased antibody responses in HSM-supplemented goats. Increased expression in PBMC but no observed increase in serum concentration with 30% treatment may be speculated as some inhibitory mechanisms operating at the translational level.

IL-6 expression in PBMC and serum concentration showed similar increasing trends with increasing levels of HSM supplementation. IL-6 gene expression was slightly but nonsignificantly upregulated compared to the control diet, while serum concentrations also increased slightly (*p* < 0.05). With respect to expression of IL-6, none of the differences were significant. IL-6 is considered both a pro- and anti-inflammatory cytokine produced by a variety of cells in the body. It enhances production of acute phase proteins during infections and inflammation. We speculate there would be no significant effect of this minor increase in IL-6 due to HSM supplementation on the immune responses of goats.

TNF-α expression was not affected by HSM supplementation at 10% and 20% levels compared to that of the control group. Only 30% treatment resulted in significantly higher level as compared to that of all other treatments (*p* < 0.05). However, serum concentrations of TNF-α decreased linearly with the increasing level of HSM supplementation (*p* < 0.01). Whether the observed increase in expression with 30% level of supplementation is just an aberrant result or high levels of supplementation causes this increase needs further investigation. TNF-α is considered a proinflammatory cytokine mainly produced by activated macrophages, T lymphocytes, and natural killer (NK) cells. While optimum levels of this cytokine may be needed in immunity against infections and cancer, increased levels may be detrimental to the general health and is implicated in inflammatory and autoimmune diseases. Whether the observed decrease in production parameters of goats supplemented with 30% HSM is at least partly due to increased expression of TNF-α needs further investigation.

## 5. Conclusions

Gene expression results signify an increase in mRNA, however, no increase in protein expression demonstrates that there is some inhibition at the translational level. With HSM supplementation, there appears to be some beneficial enhancement of cell-mediated immune responses, which could be advantageous to goats that may encounter viral and other intracellular pathogens. Supplementation did not demonstrate any significant effect on antibody responses as well as serum concentration of IL-4, suggesting no influences on Th2 responses. An increase was observed in proinflammatory cytokines, IL-6, and IL-2, while a decrease was observed in TNF-α. Expression of IL-4, an anti-inflammatory cytokine, also decreased with the increasing level of supplementation, which partly explains the reason for the observed decrease in serum anti-cOVA antibody levels. Additionally, serum cortisol concentration did not demonstrate any significant differences, suggesting that HSM supplementation had no effect on cortisol production. HSM could potentially be included in growing meat goat diets at a rate of 10–20% without having any negative effects while improving cell mediated immunity. 30% supplementation caused an increase in IL-6, which is proinflammatory and could potentially result in a loss in animal performance.

## Figures and Tables

**Figure 1 animals-11-02764-f001:**
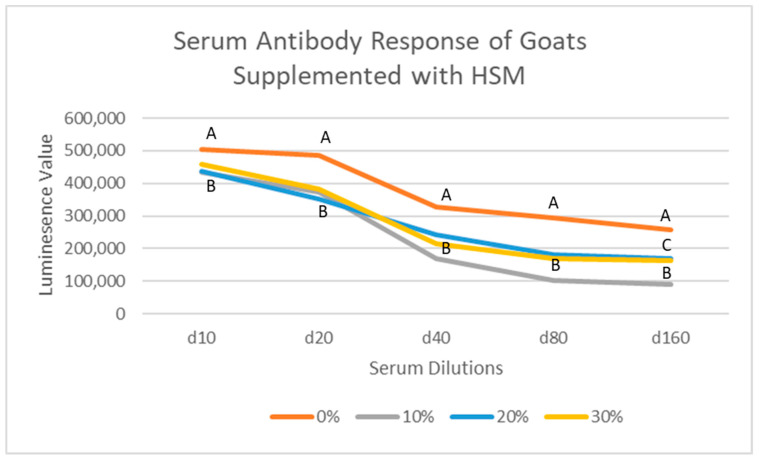
Serum antibody response of goats supplemented with varying levels HSM on day 60 of feeding period. Letters A, B, and C denote statistical significance between treatments at respective dilutions.

**Figure 2 animals-11-02764-f002:**
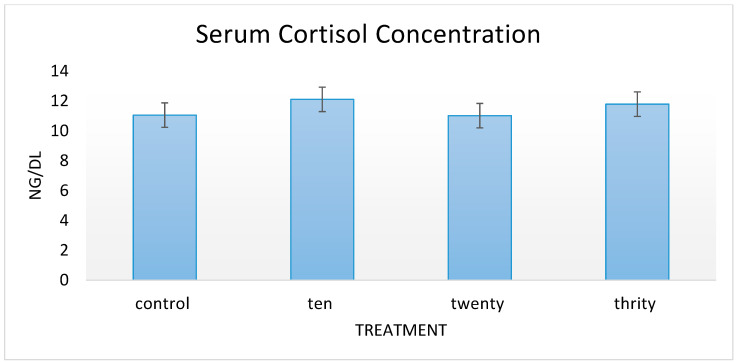
Serum cortisol concentration on day 60 of feeding period.

**Table 1 animals-11-02764-t001:** Diet composition of experimental diets utilized in 60-day feeding period.

Diets Composition for Hemp Seed Meal Experiment				
Ingredients, % of Diet	HSM Supplementation, %			
	0	10	20	30
Timothy Hay Pellet	50	50	50	50
Cracked Corn	27	22.5	17.5	11.5
Soybean Meal	19.5	14	9	5
Hemp Seed Meal	0	10	20	30
Molasses	2.5	2.5	2.5	2.5
Goat premix	1	1	1	1

Treatments are presented on an as-fed basis.

**Table 2 animals-11-02764-t002:** Primer sequences designed to determine relative quantification of select cytokine genes.

Gene	Primers (5′-3′) *	GC	Tm
*IL2-R*	CACAATGTTAAAATGCCCTTCC	40.9	53
*IL2-F*	TCCAAGCAAAAACCTGAACACC	45.5	55.6
*IL6-R*	AGTGTTTGTGGCTGGAGTGG	55	52.3
*IL6-F*	TGGATGCTTCCAATCTGGG	52.6	53
*TNF-α-R*	GATGACCTGAGTGTCTGAACC	52.4	48
*TNF-α-F*	ATGAGCACCAAAAGCATGATCC	45.5	55.5
*IL4-R*	TGAGATTCCTGTCAAGTCC	47.4	43.8
*IL4-F*	CTGATTCCAGCGCTGGTCTGC	61.9	59.4

* Primers were designed for the present study.

**Table 3 animals-11-02764-t003:** Chemical composition of experimental diets and HSM used in experimental diets fed over 60-day feeding period to growing meat goats.

Nutrient Analysis	HSM Supplementation, %				HSM
	0	10	20	30	
Dry Matter, %	89.06	88.66	89.1	89.86	89.61
Crude Protein, %	19.18	19.91	19.25	20.39	36.42
Lignin, %	3.34	4.77	6.21	7.02	12.26
ADF, %	21.06	24.67	28.96	30.97	36.47
NDF, %	33.29	35.18	39.63	42.80	49.47
Acid Hydrolysis Fat, %	3.19	3.32	4.22	4.50	11.53
Calcium, %	0.95	0.92	0.88	0.82	0.23
Phosphorus, %	0.39	0.41	0.48	0.52	1.03
Magnesium, %	0.23	0.24	0.26	0.28	0.49
Potassium, %	1.43	1.29	1.29	1.24	1.03
Sulfur, %	0.22	0.22	0.22	0.22	0.16
Sodium, %	0.09	0.08	0.09	0.09	0.03
Copper, ppm	20	15	16	22	37
Manganese, ppm	64	75	82	91	88
Zinc, ppm	71	67	74	80	76
Iron, ppm	144	154	154	137	120

All values are presented on a dry matter basis, except dry matter; NDF: neutral detergent fiber; ADF: acid detergent fiber.

**Table 4 animals-11-02764-t004:** Fatty acid composition of respective HSM diets utilized for 60-day feeding trial and pure hemp seed meal utilized in formulating diets.

Parameter, %	Pure HSM	Treatment, %			
		0	10	20	30
C:16:0 Palmitic Acid	0.77	0.45	0.44	0.48	0.47
C18:0 Stearic Acid	0.22	0.09	0.08	0.09	0.1
C18:1 Vaccenic Acid	0.09	<0.03	<0.03	0.03	0.04
C18:1 Omega 9 Oleic Acid	0.87	0.47	0.45	0.48	0.46
C18:1 Total Oleic Acid + isomers	0.99	0.52	0.49	0.54	0.53
C18:2 Linoleic Acid	3.43	0.84	1.01	1.19	1.42
C18:2 Total Linoleic Acid + isomers	3.46	0.88	1.04	1.22	1.45
C18:3 Alpha Linolenic Acid	0.92	0.17	0.25	0.32	0.42
C18:3 Gamma Linolenic Acid	0.34	<0.02	0.03	0.06	0.1
C18:3, Total Linolenic Acid + isomers	1.26	<0.02	0.29	0.38	0.51
C18:4 Octadecatetraenoic Acid	0.08	<0.02	<0.02	<0.02	0.02
C18:4 Total Octadecatetraenoic Acid	0.08	<0.02	<0.02	<0.02	0.02
C20:0 Arachidic Acid	0.07	<0.02	<0.02	0.02	0.03
C20:1 Gondoic Acid	0.03	<0.02	<0.02	<0.02	<0.02
C20:1 Total Gondoic Acid + isomers	0.05	<0.02	<0.02	<0.02	0.02
C22:0 Behenic Acid	0.04	<0.02	0.02	0.02	0.03
C24:0 Lignoceric Acid	0.03	<0.02	<0.02	<0.02	<0.02
Total Omega 3 Isomers	1.01	0.18	0.27	0.34	0.46
Total Omega 6 Isomers	3.79	0.85	1.05	1.25	1.52
Total Omega 7 Isomers	0.1	<0.05	<0.05	<0.05	<0.05
Total Omega 9 Isomers	0.92	0.48	0.46	0.49	0.48
Total Monounsaturated Fatty Acids	1.06	0.55	0.53	0.57	0.56
Total Polyunsaturated Fatty Acids	4.82	1.05	1.34	1.61	1.99
Total Saturated Fatty Acids	1.17	0.63	0.62	0.68	0.68
Total Trans Fatty Acids	0.03	0.04	0.03	0.05	0.05
Total Fat as Triglycerides	7.39	2.38	2.64	3.04	3.43
Total Fatty Acids	7.07	2.28	2.52	2.91	3.28

**Table 5 animals-11-02764-t005:** White blood cell counts on day 60 of feeding period.

Parameter	Treatment (%)				SEM	*p*-Value ^1^	
	0	10	20	30		Linear	Quadratic
White Blood Cells, K/µL	17.2	18.6	18.6	18.2	3.31	0.57	0.42
Neutrophils (%)	45.0	44.2	41.8	42.5	21.01	0.42	0.78
Lymphocytes (%)	48.5	49.9	52.3	51.0	26.68	0.38	0.59
Monocytes (%)	5.2	4.7	4.8	5.6	1.39	0.65	0.42
Eosinophils (%)	1.0	0.9	0.6	1.2	0.47	0.92	0.45
Basophils (%)	0.4	0.3	0.3	1.2	1.86	0.23	0.27
Platelets (K/µL)	713.6	735.6	669.1	600.1	225.80	0.21	0.22

^1^*p*-value from results of an orthogonal contrast for equally spaced treatments.

**Table 6 animals-11-02764-t006:** Skin fold thickness after PHA challenge and relative quantification for select cytokines.

Skin Fold Thickness					
Parameters	Treatments (%)				SEM
	0%	10%	20%	30%	
24 h Skinfold Thickness, mm	9.5 ^a^	16.45 ^b^	15.75 ^b^	15.6 ^b^	1.228

^a,b^ Means with different superscripts differ significantly (*p* < 0.05).

**Table 7 animals-11-02764-t007:** Least square mean values for serum concentration of select cytokines on day 60 of feeding trial.

Parameter	Treatments (%)				SEM	*p*-Value ^1^	
Cytokine Concentration	0	10	20	30		Linear	Quadratic
*IL-2* (pg/mL)	382.10	514.36	347.23	617.96	59.227	0.08	0.30
*IL-4* (pg/mL)	213.68	190.84	153.75	187.06	200.687	0.18	0.16
*IL-6* (pg/mL)	45.63	52.66	55.48	69.15	20.259	0.02	0.61
*TNF-α*	314.34	164.24	105.19	132.66	159.192	0.01	0.10

^1^*p*-values established as the results of a polynomial contrast for equally spaced treatments.

**Table 8 animals-11-02764-t008:** Relative quantification values of the respective cytokines in growing meat goats supplemented with varying levels of HSM on day 60 of feeding period.

Cytokine Expression					
Parameters	Treatment, %				SEM
	0	10	20	30	
*IL-2, 2* ^−ΔΔ*ct*^	1	0.31	23.1	23.7	7.340
*IL-4, 2* ^−ΔΔ*ct*^	1	0.3	0.8	10.8 ^a^	1.76
*IL-6, 2* ^−ΔΔ*ct*^	1	1.2	1.0	4.8	1.06
*TNF-α, 2* ^−ΔΔ*ct*^	1	0.5	2.1	10.5 ^a^	1.696

^a^ Values in the same row with different superscripts are significantly different.

## Data Availability

Data available on request due to restrictions. The data presented in this study are available on request from the corresponding author. The data are not publicly available due to the nature of this work.

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
