# Peer review of "Effect of Varying Levels of Hempseed Meal Supplementation on Humoral and Cell-Mediated Immune Responses of Goats"

_animals, 2021, doi:10.3390/ani11102764_

Round 1

Reviewer 1 Report

The abstract very well written with a flaw that prevents us from issuing a more accurate opinion. Treatments were not mentioned.

A table was not presented with the composition of the foods used, mainly the hempseed meal, as well as the diets.
Diets were supposed to be isonitrogen, but they weren't. The contents of CP, calcium, fiber, fat, iron and sodium, however, there were many changes, according to the authors' own report.
These changes could certainly influence the variables evaluated, which leads us to reject the paper

Author Response

  • Reviewer’s 1 Report:
    • The abstract very well written with a flaw that prevents us from issuing a more accurate opinion. Treatments were not mentioned.
    • A table was not presented with the composition of the foods used, mainly the hempseed meal, as well as the diets. Diets were supposed to be isonitrogen, but they weren't. The contents of CP, calcium, fiber, fat, iron and sodium, however, there were many changes, according to the authors' own report. These changes could certainly influence the variables evaluated, which leads us to reject the paper.
    • Response:
      • A table has been added
      • Diets were meant to be iso-nitrogenous; however, there will always be some variability when working with agricultural by-products. HSM is high in fiber, fat, and protein, which could make it an ideal feed for ruminants. This experiment was designed to assess the way producers would actually utilize this product in their production scenario (i.e., 0, 10, 20, and 30% supplementation). Additionally, a goat mineral premix was incorporated at a rate of 1% in all diets to ensure the animals needs were met.

Reviewer 2 Report

Some spelling is not so standard. For example, “Control group” in line 36 , “Βeta-Actin” in line 129
and “Tumor necrosis factor-α” in line 368; And “COVA” appears in line 164 is not uniformed with
“cOVA” in line 217 and others.
1) Please uniform the styles of subtitles. For example, “4.2. White Blood Cell Counts” “4.3.
Intradermal skinfold thickness following PHA Challenge”.
2) Please confirm that all the “SEM” values are correct. For example, “SEM” values in table
3.
3) Maybe it is better to express “ten, twenty, thirty” in figures 2 and Table 6 as
“10%,20%,30%” or “10 ,20 ,30” to confirm with Table 4 or Table 5.
4) It is not uniform for references. For example, reference 16, please correct its style and all
the other references carefully before submitting the revised manuscript.

Author Response

Reviewer’s 2 Report:

  • Some spelling is not so standard. For example, “Control group” in line 36 , “Βeta-Actin” in line 129 and “Tumor necrosis factor-α” in line 368; And “COVA” appears in line 164 is not uniformed with “cOVA” in line 217 and others.
    • Please uniform the styles of subtitles. For example, “4.2. White Blood Cell Counts” “4.3. Intradermal skinfold thickness following PHA Challenge”.
    • Please confirm that all the “SEM” values are correct. For example, “SEM” values in table 3.
    • Maybe it is better to express “ten, twenty, thirty” in figures 2 and Table 6 as “10%,20%,30%” or “10 ,20 ,30” to confirm with Table 4 or Table 5.
    • It is not uniform for references. For example, reference 16, please correct its style and all the other references carefully before submitting the revised manuscript.
    • Response:
      • The not so standard spellings have been corrected (control group, beta-actin, tumor necrosis factor)
      • The subtitle styles have been corrected. Additionally, all tables and figures have been updated to the format of the style in the latest publication.
      • All SEM values have been double-checked.
      • All tables have been adjusted in the treatment section to match each other.
      • The endnote template for the references has been updated to that required by MDPI.

Reviewer 3 Report

Present manuscript is a good examples of good manuscripts. I have only two questions. Do authors think that with a second challenge of COVA, the results could be different?. Complement system is very important in immunity, do you still have some serum to perform complement systems assays?, this will give the manuscript an extra value. 

Author Response

  • Reviewer’s 3 Report:
    • Present manuscript is a good examples of good manuscripts. I have only two questions. Do authors think that with a second challenge of COVA, the results could be different?. Complement system is very important in immunity, do you still have some serum to perform complement systems assays?, this will give the manuscript an extra value.
      • Response:
        • Yes, a second cOVA challenge would have resulted in higher Anti-cOVA IgG antibodies. In future studies, another round of cOVA injections will be administered. Unfortunately, we did not collect enough serum to do a complement analysis but that would make the paper richer.

Reviewer 4 Report

The manuscript entitled "Effect of Varying Levels of Hempseed Meal Supplementation on Humoral and Cell-Mediated Immune Responses of goats" is a well written manuscript and I found merit but somemodifications are required. The main criticism is regarding the discussion section in which the authors failed to stress the results obtained and to find convincing arguments to explain their results. The authors should address this comments in their revision of the manuscript.

Minor comments:

-Gross composition of the diet was not reported in Table 2 as written and must be reported in a new Table;

-Please add SEM in Figure 1;

-L317 and throughout the discussion section do not report significance of data; it is not a result section, the authors should explain their results and comment them, not reporting significance of data;

- The discussion section should be modified by adding relevant comments able to explain the results obtained and to stress the results also from a practical poin to view.

Author Response

  • Reviewer’s 4 Report:
    • The manuscript entitled "Effect of Varying Levels of Hempseed Meal Supplementation on Humoral and Cell-Mediated Immune Responses of goats" is a well written manuscript and I found merit but some modifications are required. The main criticism is regarding the discussion section in which the authors failed to stress the results obtained and to find convincing arguments to explain their results. The authors should address this comments in their revision of the manuscript.
    • Minor comments:
      • Gross composition of the diet was not reported in Table 2 as written and must be reported in a new Table;
      • Please add SEM in Figure 1;
      • L317 and throughout the discussion section do not report significance of data; it is not a result section, the authors should explain their results and comment them, not reporting significance of data;
      • The discussion section should be modified by adding relevant comments able to explain the results obtained and to stress the results also from a practical point to view.
      • Response:
        • Gross Composition of the diet was added
        • If SEM was added to figure 1 they would all overlap and it would be hard to understand, so the authors choose not to add it
        • Discussion complaint: HSM is a novel feedstuff and very little work has been conducted around the world so finding data to compare it too was extremely difficult. The author tried to present the results in detail while trying to find relevant published data to compare it too.
        • SEM complaint (figure 1): These values were composite samples making a data analysis impossible so there is no SEM value.

Round 2

Reviewer 1 Report

authors answer:

Diets were meant to be iso-nitrogenous; however, there will always be some variability when working with agricultural by-products. HSM is high in fiber, fat, and protein, which could make it an ideal feed for ruminants. 

The authors' own answer gives us support to continue rejecting the paper. I agree that it is difficult to work with by-products, due to the variability in its composition, and the correct way to work with this type of food is to monitor its composition throughout the execution of the experiment.

Author Response

Thanks for your comments. If you look at each of the diets, there was a slight elevation (~1%). We pelleted the diet at a feed mill to ensure the goats would not select out more preferable parts of the diet, which resulted in a more accurate experiment. We do not have access to this feed mill but once a year so there was no way to monitor this throughout the duration of the experiment. Additionally, the nutrient composition was a representative sample from the entire trial as a sample was collected from every 5th We designed this experiment with a practical production situtation in mind. If approved as a feed producers will not balance for anything but CP. Additionally, they will include it at easy to include rates 0, 10, 20, and 30% of a diet. This type of work is not grounds for rejection as it is practical and the way the product would actually be implemented in the real world.

Reviewer 4 Report

The authors addressed my previous comments.

Author Response

Thanks so much for your comments.